# Correlation Analysis of Microbial Community Changes and Physicochemical Characteristics in Aged Vinegar Brewing

**DOI:** 10.3390/foods12183430

**Published:** 2023-09-14

**Authors:** Zhixing Hou, Jinhua Zhang, Ling Dang, Hugui Xue, Min Chen, Baoqing Bai, Yukun Yang, Tao Bo, Sanhong Fan

**Affiliations:** 1College of Life Science, Shanxi University, Taiyuan 030006, China; 18536208226@163.com (Z.H.); ever840605@sxu.edu.cn (J.Z.); x1558128058@163.com (H.X.); baoqingbai@sxu.edu.cn (B.B.); yangyukun@sxu.edu.cn (Y.Y.); 2Shanxi Key Laboratory for Research and Development of Regional Plants, Shanxi University, Taiyuan 030006, China; 3School of Health Management, Shanxi Technology and Business College, Taiyuan 030006, China; dangling1000@163.com; 4Shanxi Food Research Institute Co., Ltd., Taiyuan 030024, China; chenmin768@126.com; 5Institute of Biotechnology, Shanxi University, Taiyuan 030006, China; botao@sxu.edu.cn

**Keywords:** aged vinegar, correlation, microbial community, physicochemical characteristics

## Abstract

This study aimed to explore key physicochemical characteristics and evolutionary patterns of microbial community structure during the fermentation of aged vinegar. The correlation between microorganisms and physicochemical characteristics during fermentation was examined. The results revealed significant differences in genera at different stages of fermentation. The dominant bacteria in R1 were *Bacillus*, *Lactobacillus*, *Aspergillus*, and *Issatchenkia*. During the R2 fermentation stage, *Lactobacillus*, *Acetobacter*, and *Saccharomyces* exhibited an upward trend and finally became the dominant bacteria. Aspergillus was the main bacterial genus at the end of overall fermentation. The correlation analysis showed that the bacterial genera significantly positively and negatively correlated with reducing sugars and amino acid nitrogen were the same in Cuqu. Similarly, the bacterial genera significantly positively and negatively correlated with pH and saccharification power were the same. pH, reducing sugar, and saccharification ability were mainly positively correlated with bacterial genera during fermentation. Further, studies found that the overall correlation between fungal communities and physicochemical characteristics was weaker than the correlation with bacteria during fermentation.

## 1. Introduction

Domestic vinegar is mainly grain vinegar. It is made by using grains as raw materials and introducing various microorganisms and enzymes by adding Cuqu through a complex fermentation process. Cuqu is a natural fermentation agent derived from barley, wheat, bran, peas, and other raw materials following a specific formula. These materials are crushed, mixed with water, and then compressed into Cuqu [1]. Vinegar mash, however, is formed through the fermentation of raw grains by adding Cuqu to the grains. The quality of Cuqu and vinegar mash plays a decisive role in the final quality of aged vinegar products. The brewing of vinegar mainly involves two major fermentation processes: alcohol fermentation and acetic acid fermentation. Saccharification enzymes produced by yeasts, molds, and other bacteria play a key role in the saccharification and alcoholization of raw materials during the alcoholic fermentation stage. The acetic acid fermentation stage mainly metabolizes alcohol into acetic acid through bacterial strains such as acetic acid bacteria and lactic acid bacteria [2]. Accurately grasping and understanding the changes in physicochemical characteristics during the fermentation process of aged vinegar is essential for reasonably controlling the production process of aged vinegar, for optimization, and for the improvement of the production process, and also to provide certain theoretical support for the transformation of the aged vinegar industry to mechanized production. Therefore, this study tracked and sampled vinegar mash during the brewing of aged vinegar and examined the dynamic changes in physicochemical characteristics such as moisture content, pH value, and total acid content during fermentation.

Vinegar brewing is an open and complex fermentation process involving a large number of microorganisms. These microorganisms continuously evolve and succeed, playing an essential role in the quality of aged vinegar [2]. Therefore, studying the dynamic changes in microorganisms during vinegar brewing is of great significance for improving the quality of vinegar. At present, the study of microorganisms in the brewing of aged vinegar via isolation and culture [3] and polymerase chain reaction–denaturing gradient gel electrophoresis (PCR-DGGE) [4] has been more mature. However, these methods have certain shortcomings; for instance, isolation and culture methods cannot isolate all microorganisms in the sample because the culture environment is not the same as the actual environment in which the microorganisms live [4]. PCR-DGGE suffers from low resolution and only a few recognized species [5]. With the popularization and application of high-throughput sequencing technology, PCR-DGGE has not only improved the coverage and efficiency of studying microbial community composition [6], but also significantly broadened the understanding of microbial community composition and succession. The application of high-throughput sequencing technology in aged vinegar has also been reported. Zhu et al. [7] applied high-throughput sequencing technology to study the microbial changes during the acetic acid fermentation stage of the brewing process, and Nie et al. [8] systematically explored the bacterial and fungal diversity of aged vinegar throughout the brewing stage. These studies played a significant role in understanding microbial community changes during the brewing of aged vinegar.

Compared with previous studies, this study selected Cuqu and vinegar mash for use during the fermentation of Shanxi aged vinegar as the research object. High-throughput sequencing technology was used to systematically study the dynamic changes in bacterial and fungal communities during the fermentation of Shanxi aged vinegar. At the same time, the differences in genera between the alcohol fermentation stage and the acetic acid fermentation stage were also explored. The correlation between microorganisms and physicochemical properties during the brewing of Shanxi aged vinegar was analyzed for the first time. This study also provided a theoretical basis for further studies to establish a corresponding rapid test method and microbial standard system to examine any abnormality in the fermentation of aged vinegar.

## 2. Materials and Methods

### 2.1. Reagents and Materials

Sodium hydroxide, formaldehyde, DNS reagent, acetic acid, and sodium acetate were purchased from Solebao (Beijing, China). Glucose and casein were purchased from Tianjin Guangfu Technology Development Co., Ltd. (Tianjin, China) NaH_2_PO_4_ and Na_2_HPO_4_ were purchased from Tianjin Bodi Chemical Co., Ltd. (Tianjin, China) A GeneJET gel recycling kit was purchased from Thermo Fisher Scientific. A CTAB Genomic DNA extraction kit and DPPH were purchased from Beijing Baiao Leibo Technology Co., Ltd. (Beijing, China).

### 2.2. Instruments and Equipment

A pH meter was sourced from Shanghai Yuejin Medical Instrument Co., Ltd. (Shanghai, China); an ultraviolet spectrophotometer from Shanghai Spectral Instrument Co., Ltd. (Shanghai, China); a moisture content analyzer from Shenzhen Guanya Moisture Instrument Technology Co., Ltd. (Shenzhen, China); an H2100R desktop high-speed frozen centrifuge from Hunan Xiangyi Laboratory Instrument Development Co., Ltd. (Changsha, China); an MX-S vortex mixer from Beijing Jiahang Bochuang Technology Co., Ltd. (Beijing, China); a NanoBio 200 ultra-micro spectrophotometer from Aopu Tiancheng Technology Co., Ltd. (Shanghai, China); a gel imaging system from Beijing Boao Jingdian Biotechnology Co., Ltd. (Beijing, China); and a T100 PCR amplification instrument from Bio-Rad, Hercules, CA, USA.

### 2.3. Experimental Methods

#### 2.3.1. Sample

From April to May 2021, samples of Cuqu and vinegar mash used in the same production batch were collected from a vinegar factory in Yangquan City, Shanxi Province.

For the Cuqu sample (C0), the Cuqu blocks used in production were first crushed. A 5-point sampling method was then used. After thorough mixing, 500 g of the sample was collected and placed in a sterile sealed bag. This bag was stored in a refrigerator at −80 °C for future use.

Vinegar mash samples were collected on the following days of fermentation: 1st day (C1), 3rd day (C2), 5th day (C3), 7th day (C4), 10th day (C5), 13th day (C6), 16th day (C7), 20th day (C8), and 25th day (C9) of fermentation. The sampling point was located 35 cm below the surface of the vinegar mash. Further, 500 g of samples were extracted and placed in a sterile sealed bag. These bags were stored in a refrigerator at −80 °C for future use.

#### 2.3.2. Method for Determining Physicochemical Characteristics

In addition to measuring the moisture content, for the assessment of remaining physicochemical characteristics, 10 g of the sample was taken and immersed in 30 mL of distilled water for 1 h. Then, the mixture was filtered using filter paper, and the filtrate was collected for subsequent measurements.

Furthermore, 2 g samples of curd and vinegar mash were weighed, and a water content analyzer was used to assess their water content. The pH, total acid, reducing sugar, saccharification power, and amino acid nitrogen were determined as described in Ref. [9]. The calculation of the amount of soluble salt-free solids followed the brewing soy sauce process outlined in GB/T 18186-2000. Each sample underwent three parallel measurements.

#### 2.3.3. Methods for Determining Microbial Community Dynamics

The sample pretreatment method was based on a previous study [10], with three parallel treatments set for each sample.

The CTAB method was used to extract the total DNA of the sample, and the total DNA obtained was detected via electrophoresis using 1.0% agarose gel. The agarose gel electrophoresis detection parameters were as follows: marker sample loading, 2 μL; sample loading volume, 3 μL. The electrophoresis lasted for 40 min, with 1.0% agarose and a voltage of 100 V.

After testing the purity and concentration of DNA, using genomic DNA as a template, PCR amplification was performed using primers 341F (5′-GCTACGGGNGGWGCAG-3′)/805R (5′-GATACHVGGGTATCTAATCC-3′) and 1737F (5′-GAAGTAAAAGTCGTAACAG-3′)/2043R (5′-GCTGTGTTCATCGATGC-3′) to amplify the gene sequences of bacterial 16S rDNA V3–V4 region and fungal ITS1 region, respectively. The PCR amplification system had a volume of 30 μL, consisting of the following components: 15 µL of Phusion Master Mix (2×), 3 µL of primer (2 µmol/L), 10 µL of gDNA (1 ng/µL), and 2 µL of ddH_2_O. The reaction procedure involved the following steps: pre-denaturation at 98 °C for 1 min, followed by 30 cycles of denaturation at 98 °C for 10 s, annealing at 50 °C for 30 s, and extension at 72 °C for 30 s; there was a final extension at 72 °C for 5 min. After amplification, 2% agarose gel electrophoresis was used for detection, and the product was retrieved using the GeneJET gel recovery kit which form Beijing Jiahang Bochuang Technology Co., Ltd. (Beijing, China). The qualified amplification products underwent high-throughput sequencing on an Illumina NovaSeq 6000 platform of Beijing Nuohe Zhiyuan Technology Co., Ltd. (Beijing, China).

### 2.4. Data Analysis

First, the barcode and primer sequences were removed from the offline data. Then, the splicing of double-ended sequencing reads was carried out using FLASH (V1.2.11) software. Fastp (V0.23.0) software was used for quality control. Usearch (10.0.259) software was used to remove chimeras. Operational Taxonomic Unit (OTU) partitioning, species annotation, and alpha diversity analysis were performed using QIIME2 (version 2021.8) software. *t* tests were conducted using R software (3.6.3). SPSS (IBM23) software was used to calculate Spearman correlation coefficients. Significance analysis of differences and plotting were conducted using the Origin8.5 software.

## 3. Results and Analyses

### 3.1. Analysis of Physicochemical Characteristics

#### 3.1.1. Analysis of the Physicochemical Characteristics of Cuqu

Table 1 depicts that the moisture content of the Cuqu used for the fermentation of Shanxi aged vinegar was 13.71% ± 0.36%. The moisture content of the Cuqu was generally lower than 14%, meeting the standard for the fermentation of Shanxi aged vinegar. Excessive moisture content of Cuqu can easily lead to secondary mold formation and affect the quality of Cuqu [11]. Acidity is also one of the key indicators for evaluating the quality of Daqu. The acidity of Cuqu used in this experiment was 1.49 ± 0.11 g/100 g. Organic acids in vinegar, such as tartaric acid, contribute significantly to the overall sensory flavor [12]. The amino acid nitrogen content affects the aroma of Cuqu, and the higher the content, the stronger the flavor [13]. The experimental determination showed that the amino acid and nitrogen content in Shanxi aged vinegar was 0.42 ± 0.05 g/100 g. Rong et al. found that the amino acid nitrogen content of the Daqu used in Shanxi aged vinegar was 0.16 g/100 g [14]. It was hypothesized that this experimental vinegar Cuqu had a stronger aroma than the Cuqu used in Shanxi old vinegar. Amino acids not only endow brewing wine with various taste characteristics, such as freshness, sweetness, bitterness, and astringency, but also provide nitrogen sources for the growth and metabolism of microorganisms such as yeast during fermentation, thereby promoting fermentation [15,16]. The saccharification power of Cuqu measured in this experiment was 1811.30 ± 2.31 U/g, which was higher than the saccharification power of Baoning vinegar’s “traditional Chinese medicine Cuqu” to date. It was also hypothesized that the number of mold microorganisms in Cuqu in this experiment was higher than that of “traditional Chinese medicine Cuqu” in Baoning vinegar.

#### 3.1.2. Analysis of Changes in Physicochemical Characteristics during the Fermentation of Aged Vinegar

The changes in moisture content during the fermentation process of aged vinegar are shown in Figure 1, which remained between 55.72% ± 2.17% and 62.11% ± 1.98% throughout the entire fermentation stage. The pH and total acid levels exhibited distinct changes during fermentation; pH consistently decreased to a minimum of 3.78 ± 0.08, while total acid continued to rise from the initial 1.04 ± 0.12 g/100 g to 5.31 ± 0.20 g/100 g. During the fermentation of aged vinegar, the amino acid nitrogen content increased from the initial 0.25 ± 0.05 g/100 g to the final 0.44 ± 0.08 g/100 g. Reducing sugar showed a first increasing and then decreasing trend during the fermentation of aged vinegar. The soluble salt-free solids showed a continuous upward trend throughout the fermentation process of aged vinegar, with a final content of 7.6 ± 0.62 g/100 g, which was more than twice that of 3.7 ± 0.77 g/100 g in the C1 period. Except during the fermentation period of C6–C8, the saccharification power fluctuated slightly, showing a state of first decreasing and then increasing. The remaining fermentation periods exhibited a downward trend, decreasing from 900.61 ± 2.41 U/g at the beginning to 393.18 ± 1.96 U/g at the end of fermentation.

During the reproduction and metabolism of microorganisms, water was produced. At the same time, during the acetic acid fermentation stage, daily flipping of the vinegar mash could cause some water loss, resulting in fluctuation in the water content. During the early stage of fermentation, with sufficient raw materials, a large number of microorganisms begin to accumulate and metabolize; this included acid-producing bacteria such as acetic acid bacteria, which can oxidize ethanol, propanol, and butanol to acetic acid, pyruvate, and butyric acid [17], respectively. Therefore, pH changes were significant. C4–C8 was the acetification stage. Acetic acid bacteria metabolized a large amount of the ethanol accumulated during the ethanol fermentation stage into acetic acid, resulting in an increase in the total acid content. In addition to acetic acid, organic acids such as succinic acid, oxalic acid, and lactic acid were also produced. These acids could buffer H^+^ in acetic acid [18,19], resulting in a sharp increase in total acid content but a relatively gentle pH change. During the early stage of fermentation, the content of amino acid nitrogen significantly increased, possibly due to the decomposition of protein in the raw material by microorganisms into amino acids and peptides [20]. The yeast might self-dissolve and release peptide substances [21] during the later stage of alcohol fermentation. During the C1–C2 period, reducing sugars increased significantly, and mold decomposed a large amount of the starch in the raw material into reducing sugars [22]. Then, the consumption rate of reducing sugars was greater than the generation rate, mainly reflected in the utilization of reducing sugars by yeast to produce alcohol [23]. During the early stage of fermentation, the starch in the bran was decomposed by Cuqu and the amylase contained within the bran itself. The findings of this study aligned with the trend of the changes in saccharification capacity during the fermentation of Sichuan bran vinegar. It was speculated that the accumulation of alcohol and acid substances during fermentation inhibited microbial metabolism and led to a decrease in saccharification capacity [24].

### 3.2. Analysis of Microbial Community Changes during the Fermentation of Aged Vinegar

#### 3.2.1. Dilution Curve Analysis

Figure 2 reveals that as the sequencing depth continued to increase, the dilution curves for bacteria and fungi eventually tended to flatten. This indicated a high coverage of sequencing quantity, providing the microbial information of most bacteria and fungi in the Cuqu and vinegar mash samples. The sequencing results were considered reliable.

#### 3.2.2. Alpha Diversity Analysis

The Chao1 and Simpson indexes were used to analyze the microbial diversity during the fermentation of Shanxi aged vinegar. Table 2 presents the analysis results, especially the Simpson index. The Simpson index for bacteria fluctuated between 0.536 ± 0.151 and 0.960 ± 0.054, and for fungi between 0.539 ± 0.072 and 0.868 ± 0.011, indicating a continuous change in the diversity of fungi and bacterial communities during fermentation. These indexes highlighted differences in the diversity and richness of the microbial communities in Cuqu and vinegar mash at various stages of fermentation of Shanxi aged vinegar.

#### 3.2.3. Venn Plot Analysis of Different Fermentation Stages

Cuqu (C0) was set as group R1, the alcohol fermentation stage (C1–C4) as group R2, and the acetic acid fermentation stage (C5–C9) as group R3. Figure 3 illustrates that 3529 bacterial OTUs were identified in the Cuqu and vinegar mash samples. The number of bacterial OTUs among different stages was as follows: R1 (827) < R3 (1832) < R2 (2143), with 388 bacterial OTUs detected across the three stages. Among these, the number of bacterial OTUs in R2 reached more than twice that of R1, which might be due to the addition of additional raw materials during alcohol fermentation, thereby introducing new strains [25]. Additionally, the number of bacterial OTUs in the R2 group was the largest, indicating the most abundant types of bacteria involved during the alcohol fermentation stage. It could be due to some microorganisms gradually adapting to the fermentation environment and continuously enriching, reaching the detection threshold for high-throughput sequencing. Further, 1487 fungal OTUs were detected during the entire fermentation process, with R1 (73) < R2 (577) < R3 (1140) at different stages. The total number of fungal OTUs in the three stages was 52. R3 exhibited the highest number of OTUs.

#### 3.2.4. Analysis of the Succession of Bacterial Community Structure

Firmicutes and Proteobacteria always existed in the phylum-based bacterial community, whether in Cuqu (C0) or the vinegar mash samples (C1–C9), and the relative abundance of these two phyla in each sample reached over 85.0% ± 1.5% (Figure 4A). Among these, Firmicutes always dominated the alcohol fermentation stage (C1–C4), and the relative abundance of Firmicutes peaked during the C2 stage, reaching 96.5% ± 1.9%. Proteobacteria exhibited little overall variation during the alcohol fermentation stage. Firmicutes and Proteobacteria showed opposite trends during the acetic acid fermentation stage (C5–C9), with Firmicutes showing a decreasing trend. Although Proteobacteria showed an increasing trend and became the last dominant phylum, its relative abundance finally reached 86.8% ± 3.6%. Bacterial community succession based on genus level is shown in Figure 4B, and the dominant bacterial genera in Cuqu (C0) were *Bacillus*, *Lactobacillus*, and *Weissella*, with a relative abundance of 37.9% ± 2.6%, 23.3% ± 1.1%, and 12.3% ± 1.8%, respectively. *Weissella* showed a decreasing trend during the alcohol fermentation stage (C1–C4), with the relative abundance decreasing from the initial 14.6% ± 2.7% to 2.8% ± 0.8%, whereas *Lactobacillus* gradually became the dominant bacterial genus, with the relative abundance increasing from 42.2% ± 3.6% to 73.2% ± 2.1%. *Acetobacter*, *Bacillus*, *Streptococcus*, and *Staphylococcus* were also detected at this stage, and the overall variation in the relative abundance was small, indicating that these bacteria were in a relatively stable state at this stage. The relative abundance of *Lactobacillus* showed a continuous decreasing trend during the acetic acid fermentation stage (C5–C9), decreasing from initially 46.5% ± 3.1% to 8.6% ± 1.8%. Despite fluctuations in the *Weissella* at this stage, the overall trend decreased. The relative abundance of *Acetobacter* increased with the fermentation and finally rose to 85.4% ± 2.2% during the C9 stage, making it the dominant genus in the whole system.

The β-glucosidase produced by *Weissella* could degrade cellulose in fermented raw materials [26], and it also had the property of producing lactic acid and bacteriostatic substances. Wheat was one of the main raw materials in Shanxi aged Cuqu production, rich in starch and cellulose, creating favorable conditions for the growth and reproduction of the *Weissella* genus [27]. *Bacillus* could endow Daqu with rich enzyme systems such as α-amylases and proteases. It participated in fermentation together with acetic acid bacteria, playing a role in improving the yield of vinegar and the formation of flavor [25]. *Lactobacillus* was mainly concentrated within Daqu during fermentation, and its metabolite lactic acid improved and regulated the flavor of aged vinegar and microbial community structure [28]. During the alcoholic fermentation stage, the interior of vinegar mash, except for the surface in contact with air, was basically anaerobic, which was suitable for the growth and propagation of lactic acid bacteria. In addition, continuous ethanol fermentation created a high-ethanol environment throughout the whole fermentation system [29], which promoted the death of bacteria not resistant to this environment. *Lactobacillus* became the dominant bacteria at this stage. After the end of alcohol fermentation, the vinegar mash (starter mash) from the third day of fermentation was added to each fermentation tank. In the following days, only a small area of the mash was flipped, and the interior remained in a relatively anaerobic environment. *Lactobacillus* could still grow, leading to a short-term increase in its relative abundance. After introducing vinegar mash, extensive stir frying was carried out to ensure that the vinegar mash was loose and could pass through oxygen. At the same time, the acidity level in the fermentation broth continued to decrease, inhibiting the growth of *Lactobacillus* and reducing its relative abundance. *Acetobacter* is a nutritive bacterium. The acetic acid produced by its metabolism is the main component of organic acid in Shanxi aged vinegar. Daily flipping of the vinegar mash for aeration was beneficial for *Acetobacter*’s growth and metabolism [30]; therefore, its relative abundance gradually increased during the acetic acid fermentation stage, becoming the dominant bacteria affecting the quality of aged vinegar. *Staphylococcus* was the most common pathogenic bacterium, but it played an essential role in certain fermented foods. In fermented meat products, *Staphylococcus* produced catalase and nitrate reductase, which could improve the color of meat products, increase their special flavor, and slow down rancidity, thereby ensuring product quality [31]. The analysis results indicated that the relative abundance of *Staphylococcus* ultimately decreased to 0.8% ± 0.1% as fermentation progressed, which was extremely low. Additionally, *Acinetobacter* and *Leuconostoc* were also detected at the end of fermentation, but their relative abundance was also extremely low. Some nonpathogenic bacteria in *Acinetobacter* secreted and produced nutrients such as vitamins and phospholipids [32]. Some nonpathogenic bacteria in *Leuconostoc* metabolized and produced various metabolic substances, such as acids and alcohols [33].

#### 3.2.5. Analysis of Fungal Community Structure Succession

Figure 4C shows the succession changes in fungal community structure at the phylum level. Nine fungal phyla were mainly detected in the whole process. Ascomycota was predominant in the Cuqu and fermentation process, with the highest content in the Cuqu (C0), and the relative abundance peaked at 99.7% ± 0.9%. Basidiomycota continued to decline from 4.5% ± 0.5% to 0.7% ± 0.1% during the alcohol fermentation stage. Its relative abundance fluctuated between 1.3% ± 0.3% and 4.6% ± 0.8% during the acetic acid fermentation stage. Although the relative abundance of Mucoromycota was relatively low during fermentation, it remained stable throughout the fermentation process, indicating that this fungus could adapt to the brewing environment of aged vinegar and impact the brewing of aged vinegar. The fungal community structure based on genus level is shown in Figure 4D. In Cuqu (C0), the main genera were *Aspergillus* and *Issatchenkia*, with relative abundances of 90.2% ± 1.5% and 5.6% ± 0.5%, respectively. During the alcohol fermentation stage (C1–C4), *Issatchenkia* and *Aspergillus* were still the main dominant genera. The relative abundance of *Issatchenkia* increased from 36.7% ± 3.8% to 75.8% ± 3.0%. The relative abundance of *Aspergillus* decreased from 52.1 ± 1.3% to 18.1% ± 0.5% at this stage. During the acetic acid fermentation stage (C5–C9), *Issatchenkia* was the main genus of fungus during the C5–C7 stage. The relative abundance of *Issatchenkia* decreased during the C8 and C9 stages, and peaked at 73.4% ± 2.2% in the C5 samples. The relative abundance of *Phaeosphaeria* and *Saccharomyces* peaked at 17.6% ± 2.9% and 11.2% ± 3.6% in the C8 samples, respectively. *Aspergillus* was the main genus at the end of fermentation, with a relative abundance of 50.9% ± 1.2%. Further, some genera with relatively low abundance, such as *Millerozyma*, *Cladosporium*, *Trichosporon*, and *Candida*, were detected during the whole fermentation process.

*Issatchenkia* is a relatively important non-*Saccharomyces cerevisiae* yeast strain which is resistant to ethanol, acid, and high temperatures and produces ethanol and ethyl acetate via the metabolism [34]. The raw materials were relatively sufficient during the early stage of fermentation, making it conducive to yeast metabolism. Therefore, this genus of yeast rapidly multiplied. With the progress of fermentation, microorganisms adapted to the environment of the system and tended to be stable [35,36]. *Aspergillus* can produce saccharifying enzymes and amylases [37,38], which play a good role in saccharifying starch in raw materials. During the alcohol fermentation stage, it was related to the production of flavor substances such as isoamyl alcohol, isobutanol, and ethyl acetate [39]. The cell wall structure of molds consists of an outer layer of β-glucan, a middle layer of glycoprotein, and an inner layer of chitin, while that of yeasts consists of an inner layer of dextran, an intermediate layer composed mainly of proteins, and an outer layer of mannan. This “sandwich-type” cell wall structure can still be stabilized and not destroyed under acidic conditions [40]. Therefore, *Issatchenkia* and *Aspergilus* could still exist during the acetic acid fermentation stage. Some low-abundance fungi also play a role in fermentation. For example, *Cladosporium* can produce cellulase [41]. *Candida* has good fermentation ability and alcohol tolerance. It can reduce alcohol content with water and secrete glycosides enzymatically to hydrolyze aromatic glycosides, releasing terpenes and improving a wine’s aroma [42]. Additionally, the genus *Russula* was only detected during the acetic acid fermentation stage. In addition to being an edible fungus with multiple nutrients, the genus *Russula* is also a medicinal fungus with antitumor and antioxidant effects. In recent years, it has attracted attention due to its promising developmental potential [43].

#### 3.2.6. Beta Diversity Analysis

Beta diversity was used to characterize the compositional differences between different samples. NMDS analysis was considered to reflect the differences in microbial community structure among the different samples. The analysis results are shown in Figure 5, with bacterial and fungal stress values of 0.09 and 0.089, respectively. This indicated a significant separation among the 10 samples, suggesting significant differences in bacterial and fungal microbial communities between each sample. As shown in Figure 5A, the four samples from the alcohol fermentation stage (C1–C4) were positioned in the second and third quadrants, while the five samples from the acetic acid fermentation stage (C5–C9) were located in the first and fourth quadrants, indicating differences in bacterial community structure between the two stages. The parallel samples within the bacterial and fungal groups were also widely dispersed, possibly due to uneven sampling.

#### 3.2.7. Screening of Differential Bacterial Genera in Two Fermentation Stages

Studies exploring the potential differential genera of bacteria between the alcohol fermentation stage and the acetic acid fermentation stage are limited [7]. Hence, this study involved intergroup *t* test analysis to further identify the potentially significant differences in bacterial genera between these two major brewing stages. As shown in Figure 6, 23 genera with significant differences were found in the two major fermentation stages at the genus level, including 15 bacterial genera, such as *Acetobacter*, *Lactobacillus*, *Bacillus*, *Weissella*, and so forth, and 8 fungal genera, such as *Cladosporium*, *Acremonium*, *Saitozyma*, *Alternaria*, and so forth.

### 3.3. Correlation Analysis

#### 3.3.1. Correlation Analysis between the Physicochemical Characteristics of Cuqu and Microorganisms

The top 15 bacterial and fungal microorganisms with a relative abundance of Cuqu at the genus level were selected, and the correlation between these microorganisms and the physicochemical characteristics of Cuqu based on the Spearman correlation coefficient was explored. As shown in Figure 7, 15 bacterial and 11 fungal genera had a significant correlation with the physicochemical characteristics of Cuqu. *Acetobacter* and *Paenibacillus* were positively correlated with moisture content, but negatively correlated with soluble salt-free solids. The bacterial genera significantly positively and negatively correlated with reducing sugars and amino acid nitrogen were the same, while the bacterial genera significantly positively and negatively correlated with pH and saccharification power were the same. The fungal genera that exhibited a significant negative correlation with soluble salt-free solids, total acid, and amino acid nitrogen were *Trichosporon*, *Candida*, *Neurospora*, and *Apiotrichum*. *Issatchenkia* was only positively correlated with reducing sugar and not with other physicochemical characteristics.

#### 3.3.2. Correlation Analysis between Physicochemical Characteristics of the Fermentation Process and Microorganisms

We selected the top 15 bacterial and fungal microorganisms with relative abundance during fermentation at the genus level and explored the correlation between these microorganisms and their physicochemical characteristics based on the Spearman correlation coefficient. As illustrated in Figure 8, 11 bacterial and fungal genera significantly correlated with the physicochemical characteristics of vinegar mash. pH and saccharifying power were significantly positively correlated with 10 bacterial genera, including *Enterococcus*, *Lactobacillus*, *Staphylococcus*, *Bacillus*, and *Weissella*, and 4 fungal genera, including *Trichosporon*, *Candida*, and *Neurospora*. These 10 bacterial genera, in addition to the family Muribaculaceae, also had a significant positive correlation with reducing sugars, and these 4 fungal genera had a significant negative correlation with soluble salt-free solids, total acids, and amino acid nitrogen. *Acetobacter* was negatively correlated with soluble salt-free solids, total acid, and amino acid nitrogen, but positively correlated with pH, saccharifying power, and reducing sugar. The four fungal genera *Cladosporium*, *Malassezia*, *Acremonium*, and *Saitozyma* significantly negatively correlated with pH, saccharification capacity, and reducing sugars, but were significantly positively correlated with soluble salt-free solids, total acids, and amino acid nitrogen. A significant negative correlation was observed between moisture content and the genera *Pediococcus*, *Escherichia coli–Shigella*, and *Diutina*; no genus was found to be significantly positively correlated with moisture content. This study found that the overall correlation between fungal communities and physicochemical characteristics was weaker compared with the correlation with bacterial communities during fermentation.

This indicated that the bacterial community was the fundamental factor affecting the quality of aged vinegar fermentation, which was similar to the findings of Yang and Yanli et al. [44,45]. *Lactobacillus* and other microorganisms grew and multiplied during the fermentation stage, and could quickly utilize the reducing sugar in the vinegar mash to start fermentation, thus resulting in the decrease in reducing sugar content. The enzyme alcohol dehydrogenase accumulated during fermentation, and the content of reducing sugars decreased rapidly as they were converted into alcohol by the enzyme. At the same time, the large-scale agitation of the vinegar mash during fermentation slowed down the growth and metabolic inhibition of acid-producing bacteria such as *Lactobacillus* and *Acetobacter* due to the increase in alcohol content. Moreover, the content of amino nitrogen in the vinegar mash was high at this stage. The degradation, oxidation, or reduction of amino acids, thereby generating acidic substances, caused a rapid increase in total acid content. The change in saccharification power was mainly related to filamentous fungi. Some molds secreted large amounts of saccharolytic enzymes and proteases, resulting in a rapid increase in saccharification power. As fermentation progressed, the consumption of nutrients and the decrease in pH inhibited the growth of bacteria and molds, which led to the inhibition of saccharase secretion. Further, the acid also inhibited saccharase activity; therefore, the saccharification power decreased linearly.

## 4. Conclusions

In this study, the main physicochemical characteristics of Cuqu and aged vinegar were detected and analyzed during brewing. Additionally, the 16S rDNA V3–V4 regions of the sample bacteria and the fungal ITS1 region were detected using high-throughput sequencing technology. This helped analyze the microbial diversity and structural composition during the brewing of Shanxi aged vinegar and explore the correlation between microorganisms and physicochemical characteristics. The following conclusions were made.

In Cuqu (R1), the main bacterial genera included *Bacillus*, *Lactobacillus*, and *Weissella*, and the main fungal genera were *Aspergillus* and *Issatchenkia*. Among these, *Bacillus* and *Weissella* provided vinegar yeast with α-rich fungal strains, such as amylase and protease, playing an essential role in the formation of Cuqu aroma. During the alcohol fermentation stage (R2), *Lactobacillus* and *Weissella* showed a trend of growth and decline. During the later stage of fermentation, *Lactobacillus* gradually became the dominant bacterial genus, and *Issatchenkia* also showed an upward trend, finally becoming the predominant fungal genus. During the acetic acid fermentation stage (R3), the dominance of the bacterial genus shifted from the original *Lactobacillus* to *Acetobacter*. *Issatchenkia* was the main bacteria in the C5–C7 period; *Aspergillus* was the main bacteria at the end of fermentation. *Cladosporium*, *Candida*, and other bacteria existed in the whole fermentation process, although their relative abundance was low. Through *t* test analysis, 23 significantly different bacterial genera were found between R2 and R3, including 15 bacterial and 8 fungal genera.

The correlation analysis in Cuqu revealed that *Acetobacter* and *Paenibacillus* were significantly positively correlated with moisture content but negatively correlated with soluble salt-free solids. The bacterial genera significantly positively and negatively correlated with reducing sugars and amino acid nitrogen were the same. Also, the bacterial genera significantly positively and negatively correlated with pH and saccharification power were the same. A significant correlation was observed between 11 bacterial and 11 fungal genera and the physicochemical characteristics of vinegar mash during fermentation. pH, reducing sugar, and saccharification ability were mainly positively correlated with bacterial genera. *Acetobacter* was negatively correlated with soluble salt-free solids, total acid, and amino acid nitrogen, but positively correlated with pH, saccharifying power, and reducing sugar. The four fungal genera of *Cladosporium*, *Malassezia*, *Acremonium*, and *Saitozyma* had significant negative correlations with pH, saccharification capacity, and reducing sugar level, while they were significantly positively correlated with soluble salt-free solids, total acids, and amino acid nitrogen. A significant negative correlation was observed between water content and the genera *Pediococcus*, *Escherichia coli–Shigella*, and *Diutina*. No significant positive correlation was observed between bacterial genera and water content. In summary, compared with bacteria, the overall correlation between fungal communities and physicochemical characteristics was weak during fermentation, and bacterial communities primarily affected the physicochemical characteristics of aged vinegar.

## Figures and Tables

**Figure 1 foods-12-03430-f001:**
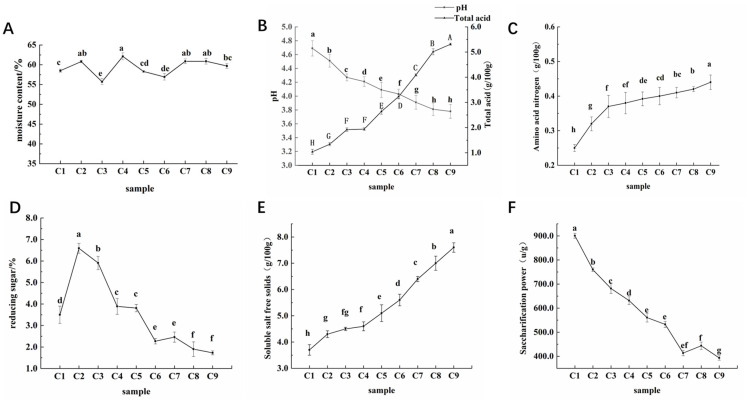
Measurement results of physicochemical parameters during the fermentation of aged vinegar: (**A**) moisture content, (**B**) pH and total acid, (**C**) amino acid nitrogen, (**D**) reducing sugar, (**E**) soluble salt-free solids, and (**F**) saccharification power. Note: The Cuqu (C0) was set as group R1, the alcoholic fermentation stage (C1–C4) as group R2, and the acetic acid fermentation stage (C5–C9) as group R3. A–H: indicates significant difference; a–h: indicates significant difference.

**Figure 2 foods-12-03430-f002:**
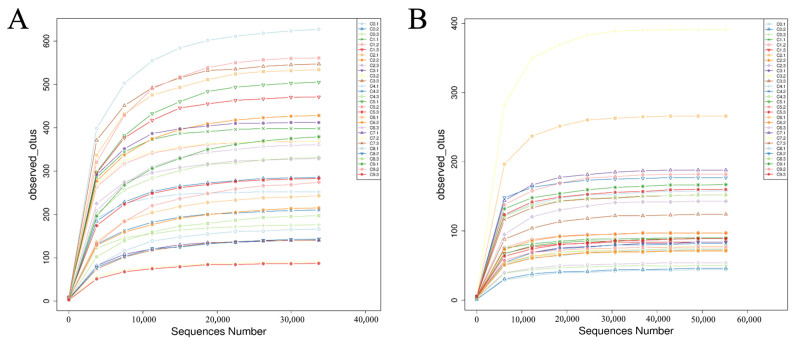
Dilution curves for (**A**) bacteria and (**B**) fungi.

**Figure 3 foods-12-03430-f003:**
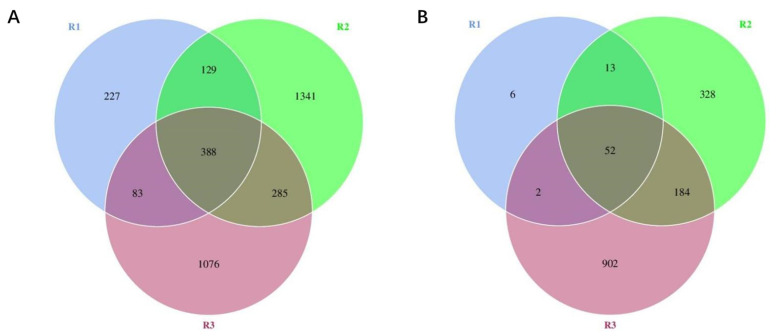
Venn diagram depicting different stages of (**A**) bacterial and (**B**) fungal diversities.

**Figure 4 foods-12-03430-f004:**
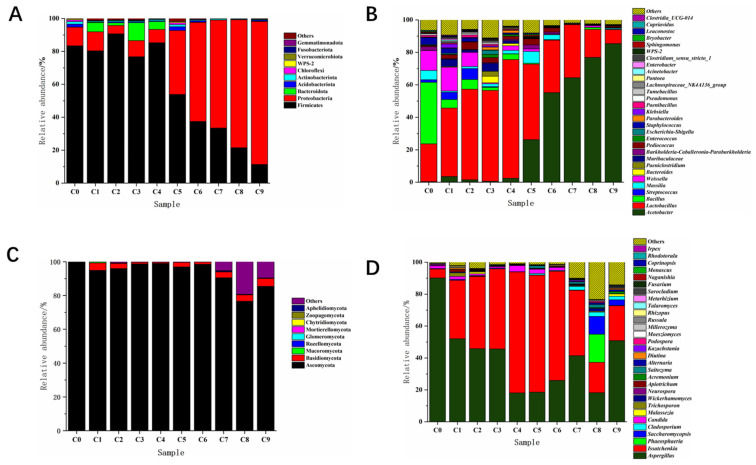
Analysis of the dynamic changes in microbial community during the fermentation of vinegar. (**A**) Dynamic analysis of the relative abundance of bacteria at the phylum level, (**B**) of bacteria at the genus level, (**C**) of fungi at the phylum level, and (**D**) of fungi at the genus level. Note: The Cuqu (C0) was set as group R1, the alcoholic fermentation stage (C1–C4) as group R2, and the acetic acid fermentation stage (C5–C9) as group R3.

**Figure 5 foods-12-03430-f005:**
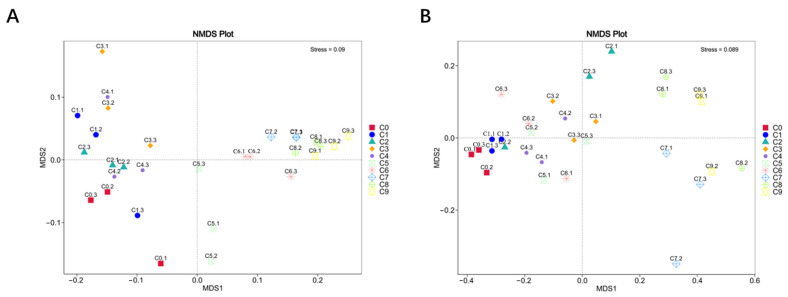
NMDS analysis of (**A**) bacteria and (**B**) fungi.

**Figure 6 foods-12-03430-f006:**
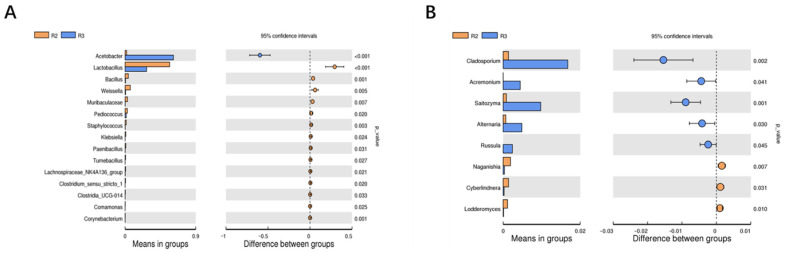
*t* test analysis of microbial community structure of (**A**) bacteria and (**B**) fungi.

**Figure 7 foods-12-03430-f007:**
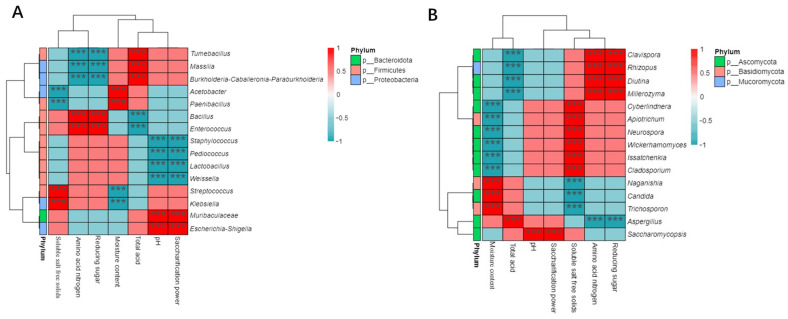
Correlation heat map analysis of physicochemical parameters of Cuqu with microorganisms: (**A**) bacteria and (**B**) fungi (*** *p* < 0.001).

**Figure 8 foods-12-03430-f008:**
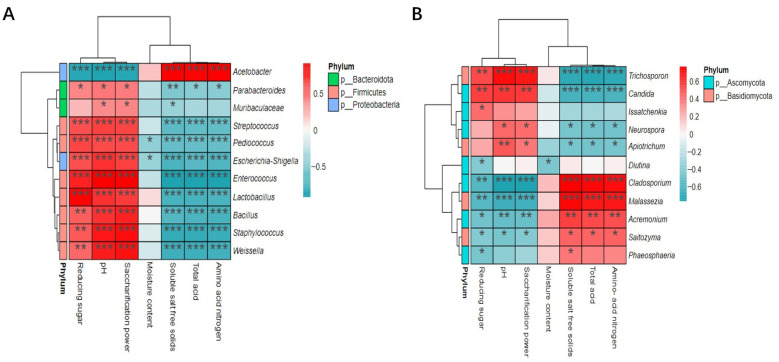
Correlation heat map analysis of physicochemical parameters and microorganisms in fermentation process: (**A**) bacterial and (**B**) fungi (*** *p* < 0.001; ** *p* < 0.01; * *p* < 0.05).

**Table 1 foods-12-03430-t001:** Results of physicochemical parameters of Cuqu.

Sample	Moisture Content (%)	pH	Total Acid (g/100 g)	Amino Acid Nitrogen (g/100 g)	Reducing Sugar (%)	Soluble Salt-Free Solids (g/100 g)	Saccharifying Power (U/g)
Cuqu (C0)	13.71 ± 0.36	5.77 ± 0.14	1.49 ± 0.11	0.42 ± 0.05	12.48 ± 0.67	3.30 ± 0.24	1811.30 ± 2.31

Note: The data are presented as the average value ± standard deviation of three replicates.

**Table 2 foods-12-03430-t002:** Bacterial and fungal alpha diversity index.

Sample	Bacteria		Fungi	
	Chao1 Index	Simpson Index	Chao1 Index	Simpson Index
C0	230.904 ± 15.567 ^g^	0.885 ± 0.092 ^e^	47.083 ± 7.091 ^j^	0.654 ± 0.077 ^f^
C1	441.878 ± 18.685 ^c^	0.960 ± 0.054 ^a^	90.333 ± 13.953 ^f^	0.752 ± 0.079 ^c^
C2	413.833 ± 17.683 ^d^	0.943 ± 0.041 ^b^	168.667 ± 11.669 ^c^	0.710 ± 0.147 ^e^
C3	444.700 ± 22.778 ^b^	0.939 ± 0.184 ^c^	96.306 ± 9.954 ^e^	0.614 ± 0.064 ^g^
C4	258.899 ± 9.078 ^f^	0.840 ± 0.121 ^f^	76.569 ± 15.553 ^h^	0.539 ± 0.072 ^j^
C5	546.000 ± 14.770 ^a^	0.914 ± 0.009 ^d^	85.778 ± 12.221 ^g^	0.602 ± 0.080 ^i^
C6	230.177 ± 24.448 ^h^	0.803 ± 0.116 ^g^	66.333 ± 9.005 ^i^	0.607 ± 0.034 ^h^
C7	137.011 ± 10.986 ^j^	0.748 ± 0.029 ^h^	170.375 ± 16.866 ^a^	0.713 ± 0.091 ^d^
C8	165.389 ± 15.007 ^i^	0.676 ± 0.116 ^i^	163.287 ± 19.099 ^d^	0.868 ± 0.011 ^a^
C9	335.181 ± 8.903 ^e^	0.536 ± 0.151 ^j^	170.017 ± 15.022 ^b^	0.801 ± 0.0663 ^b^

Note: Different letters in the same column indicate significant differences (*p* < 0.05).

## Data Availability

The data obtained in this study are available within this manuscript.

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
