# Peer review of "Correlation Analysis of Microbial Community Changes and Physicochemical Characteristics in Aged Vinegar Brewing"

_foods, 2023, doi:10.3390/foods12183430_

Round 1

Reviewer 1 Report

INTRODUCTION: IS THERE ANY PREVIOUS STUDIES ON THESE SUBJECTS (LIKE USING DIFFERENT METHODS), IF THERE IS, WHAT DIFFERENCES AND WHAT IS THE ORIGINAL VALUE OF THIS CURRENT WORK? THE INTRODUCTİON PART IS WEAK AND SHOULD BE DEVELOPED.

LINE 41: ‘OTHER FUNGAL PRODUCED BY YEAST AND MOLDS’ PLEASE CHANGE THIS STATMENT

LINE 60-63: THE SENTENCE SEEMS TOO LONG AND CONFUSİNG. PLEASE REWRITE IT BY DIVIDING IT İNTO TWO.

LINE 64-73: VERY LONG SENTENCE. PLEASE EXPRESS IT BY DIVISION BY TWO OR THREE.

LINE 74: 2.1 NO TITLE REQUIRED. THE TITLES 2.1.1 AND 2.1.2 BELOW ARE ALREADY APPROPRIATE.

LINE 104-138: NO NEED FOR TITLES LIKE 2.2.2.2 OR 2.2.3.2. PLEASE EDIT THE SUB-TITLES AGAIN.

LINE 110-114: WRITTEN FORM IS NOT SUITABLE. PLEASE DO NOT START WITH A CAPITAL LETTER AFTER THE SEMICOLON. THIS SECTION CAN BE BETTER WRITTEN

LINE 139-140: PLEASE USE PAST TENSE

LINE 154: THE FIRST LETTERS OF ORGANIC ACIDS AND TARTARIC ACIDS SHOULD BE WRITTEN IN CAPITAL, NOT CAPITAL. PLEASE REVIEW ALL TEXT FOR LOWER CASE AND CAPITAL CASTING.

LINE 150-165: THE DISCUSSION HERE IS NOT GOOD. NO COMMENT ON WHY THE RESULTS DIFFER FROM THE RESULTS IN THE REFERENCES.

LINE 264: SPECIES?

LINE 260-263: PLEASE WRITE THIS PART AGAIN

LINE 263: IS THERE AN EXPRESSION LIKE VINEGAR YEAST?

LINE 277: Weissella produces β- The characteristic of glucosidase is its ability to degrade cellu- 277 lose in fermentation materials ??

LINE 284-287: PLEASE REWRITE THIS SENTENCE.

LINE 304-308: THERE ARE PATHOGENES OF STAPYHYLOCOCCUS. BUT THERE ARE SPECİES THAT PLAY USEFUL ROLE IN FOOD FERMENTATIONS AND EVEN USED AS STARTER CULTURE. MANY NEGATIVES ARE MENTIONED IN THIS SECTION, IT IS NOT REASONABLE TO WRITE A NEGATIVE SITUATION WHILE IT IS KNOWN WHILE DISCUSSING THE RESULTS ON THE BASIS OF GENDER.

LINE 312: IT WOULD BE BETTER TO WRITE THE RESULTS OBTAINED FOR ACINETOBACTER AND LEUCONOSTOC IN ONE SENTENCE AND GO TO THE DISCUSSION AFTER THAT.

LINE 332: BACTERIA?

LINE 334: SACCHAROMYCES PLEASE.

LINE 339: Saccharomyces cerevisiae ITALIC PLEASE.

LINE 347-350: PLEASE REWRITE THIS SENTENCE.

LINE 386: THE WORD ‘INCLUDING’ CONSEQUENTLY IS USED.

LINE 412-433: THE RESULTS IN THIS SECTION SHOULD BE DISCUSSED A LOT MORE BECAUSE THIS IS ONE OF THE MOST CHALLENGE PART OF THE ARTICLE.

I DID NOT FIND THE ENGLISH OF THE MANUSCRIPT ENOUGH, IT SHOULD BE REVIEWED.

Reviewer 2 Report

This manuscript studies the microbial and fungal diversity during the vinegar production process. The technique for its study is by Illumina (independent culture method), one of the newest methods to study the microbial population with high sensitivity. Due to the high cost of this technique and the need for knowledge in bioinformatics for the analysis of the sequences, it is little used for the study of fermented foods, which makes the present work very valuable in the area.

The development of the manuscript is clear and well-founded. However, there are aspects that must be improved prior to its approval.

First, the quality of some graphs needs to be improved and in some cases the font size of the axes needs to be increased.

A diagram with the fermentation process, highlighting the sampling times (C1, C2,..., etc) and the stages R1, R2 and R3 would be beneficial.

Taking into account that the fermentation process described is for regional products that are not as well known in other regions of the world, a better explanation would be appreciated in the introduction on the raw materials and the description of solid state fermentation (unlike of the elaboration of other types of vinegars) and a description of the Cuqu and Koji, for a better understanding of the readers.

Part of the results mention the effect of the fermentation of the wheat grain, but it is not clear if it is the raw material for the vinegar studied in the present work.

Also, it would be important to highlight in the introduction if there are other studies on diversity with these techniques, or others, in this type of vinegar or in other similar fermented products.

Finally, it is the description of figure 4B, it is not clear if there is a risk of presence of pathogenic bacteria in the final product. Please clarify.

Minor fixes:

A few typos were detected, for example pH with a capital letter on line 172.

The wording of line 277 must be revised.
